# Integrating Embeddings from Multiple Protein Language Models to Improve Protein *O*-GlcNAc Site Prediction

**DOI:** 10.3390/ijms242116000

**Published:** 2023-11-06

**Authors:** Suresh Pokharel, Pawel Pratyush, Hamid D. Ismail, Junfeng Ma, Dukka B. KC

**Affiliations:** 1Department of Computer Science, Michigan Technological University, Houghton, MI 49931, USA; sureshp@mtu.edu (S.P.); ppratyus@mtu.edu (P.P.); hdismail@mtu.edu (H.D.I.); 2Department of Oncology, Lombardi Comprehensive Cancer Center, Georgetown University Medical Center, Georgetown University, Washington, DC 20057, USA; junfeng.ma@georgetown.edu

**Keywords:** *O*-GlcNAc prediction, protein language models, post-translational modification prediction, ensemble learning, embeddings

## Abstract

*O*-linked β-*N*-acetylglucosamine (*O*-GlcNAc) is a distinct monosaccharide modification of serine (S) or threonine (T) residues of nucleocytoplasmic and mitochondrial proteins. *O*-GlcNAc modification (i.e., *O*-GlcNAcylation) is involved in the regulation of diverse cellular processes, including transcription, epigenetic modifications, and cell signaling. Despite the great progress in experimentally mapping *O*-GlcNAc sites, there is an unmet need to develop robust prediction tools that can effectively locate the presence of *O*-GlcNAc sites in protein sequences of interest. In this work, we performed a comprehensive evaluation of a framework for prediction of protein *O*-GlcNAc sites using embeddings from pre-trained protein language models. In particular, we compared the performance of three protein sequence-based large protein language models (pLMs), Ankh, ESM-2, and ProtT5, for prediction of *O*-GlcNAc sites and also evaluated various ensemble strategies to integrate embeddings from these protein language models. Upon investigation, the decision-level fusion approach that integrates the decisions of the three embedding models, which we call LM-OGlcNAc-Site, outperformed the models trained on these individual language models as well as other fusion approaches and other existing predictors in almost all of the parameters evaluated. The precise prediction of *O*-GlcNAc sites will facilitate the probing of *O*-GlcNAc site-specific functions of proteins in physiology and diseases. Moreover, these findings also indicate the effectiveness of combined uses of multiple protein language models in post-translational modification prediction and open exciting avenues for further research and exploration in other protein downstream tasks. LM-OGlcNAc-Site’s web server and source code are publicly available to the community.

## 1. Introduction

Glycosylation is perhaps the most common and structurally diverse post-translational modification (PTM) of proteins [1,2]. Distinct from other glycans, *O*-linked β-*N*-acetylglucosamine (*O*-GlcNAc) is a unique intracellular monosaccharide modification [3,4]. A multi-decade research endeavor has made clear that *O*-GlcNAcylation exists in almost all kingdoms of life and even some viruses [5]. It plays important roles in almost all cellular processes examined (including genome maintenance, epigenetic regulation, protein synthesis/degradation, metabolic pathways, and signaling pathways, among others). By modulating target proteins, *O*-GlcNAc exerts various functional roles in numerous physiological and pathological events [6,7].

Protein *O*-GlcNAcylation functions in a site-specific manner. To characterize site-specific *O*-GlcNAcylation on proteins, great progress has been made, especially with the recent technological advances in high-throughput methods (e.g., mass spectrometry-based proteomics) [8,9]. So far, thousands of *O*-GlcNAc sites on myriad proteins have been catalogued [10,11,12,13]. Despite the progress, sensitive and robust techniques for global and site-specific *O*-GlcNAc analysis are still lacking. Unbiased identification of *O*-GlcNAc sites using experimental methods is also technically challenging [8,9]. Moreover, the total number of *O*-GlcNAc proteins/sites still appears to be far lower than previously predicted [14].

Prediction of *O*-GlcNAcylation sites has been difficult [15,16]. So far, several computational tools have been developed to predict *O*-GlcNAc sites. The YinOYang prediction program was based on a neural network trained on 40 experimentally determined *O*-GlcNAc acceptor sites [14]. Based on the developed dbOGAP database [10] (consisting of ~380 experimentally identified *O*-GlcNAc sites from 167 proteins), the OGlcNAcScan prediction program was developed [10]. By using almost the same positive datasets and similar negative datasets, several other machine learning models (including PGlcS [17], OGTSite [18], *O*-GlcNAcPRED [19], and *O*-GlcNAcPRED-II [20]) were developed. Although these computational methods showed great potential, their performance has still been unsatisfactory. One reason might be the very limited positive datasets used for model training and testing. Moreover, similar to mucin-type *O*-glycosylation [21], *O*-GlcNAcylation does not display a strict amino acid consensus sequence of proteins [5], making precise prediction of *O*-GlcNAc sites a very challenging task.

On the other hand, with the advancements taken from the field of natural language processing, we have witnessed the development of protein language models that learn meaningful representations of proteins in a self-supervised manner by using the vast quantity of unlabeled protein sequence databases. Some examples of these language models are Ankh [22], ESM-2 (Evolutionary Scale Modeling) [23], and ProtT5 [24]. These language models mainly differ in the transformer architectures utilized, as well as the datasets used to train these models.

Our approach here was to leverage embeddings from several pre-trained protein language models for prediction of *O*-GlcNAc sites. Witnessing recent breakthroughs in the development of large protein language models [22,23,24,25,26,27] and their effectiveness in various protein downstream tasks, including post-translational modifications [28,29,30,31,32,33,34,35,36], we explored the effectiveness of individual embeddings from three sequence-based protein language models, Ankh [22], ProtT5 [24], and ESM-2 [23], as well as the integration of embeddings from these models for the prediction of *O*-GlcNAc sites. The dataset used in this study consisted of 13900 experimentally verified *O*-GlcNAcylation sites obtained from *O*-GlcNAcAtlas [12]. In brief, three feed-forward neural networks (FFNNs) were trained independently using embeddings individually extracted from these three protein language models (Ankh [22], ESM-2 [23], and ProtT5 [24]). Moreover, with an aim to leverage the assorted strengths and diverse representations captured by these three prominent protein language models, we performed integration of the embeddings from these models using score-level and decision-level fusion. In particular, the approach that utilizes decision-level fusion of these three models performed the best, and we call this novel prediction tool LM-OGlcNAc-Site (i.e., Protein Language Model-based *O*-GlcNAc Site predictor). LM-OglcNAc-Site outperformed not only the models learned from the individual protein language models but also existing *O*-GlcNAc site predictors. We found that integrating embeddings from multiple protein language models is particularly useful for protein *O*-GlcNAc site prediction and could be a very useful approach for other bioinformatics tasks.

## 2. Results

In this section, we present the 10-fold cross-validation results obtained using various protein language models (Ankh, ESM-2, and ProtT5). Table 1 shows the comparison results of FFNN architecture using 10-fold cross-validation trained on the above-mentioned three embeddings.

Upon analyzing the table, it can be inferred that all three models produced comparable results overall. However, it is worth noting that the Ankh-FFNN model demonstrated relatively better performance in terms of the ROC-AUC. In other words, the Ankh-FFNN model appears to have achieved a higher area under the ROC curve, indicating that it has a better ability to identify positive and negative samples compared to the other models. Moreover, the other models proved their effectiveness in various metrics, implying that each model may excel at capturing distinct characteristics. The results of similar experiments performed using the class weighting cost-sensitive method are provided in Appendix A. In the subsequent sections, we will explore the performance of combining these three models.

Table 2 presents the results of the individual models in addition to the score-level fusion and decision-level fusion results of the models on the independent test set mentioned in Table 1. We call the decision-level fusion of the three pLM-FFNN models LM-OGlcNAc-Site.

According to Table 2, while the score-level fusion demonstrated comparable results, it is evident that the decision-level fusion of the three models, namely LM-OGlcNAc-Site, exhibited superior performance in terms of accuracy, sensitivity, and MCC.

Additionally, we conducted tests on the models using O-GlcNAcPRED-II’s independent test set, ensuring that none of the protein sequences in the test set were part of LM-OGlcNAc-Site’s training sequences. The results of both the individual models and the combined models in O-GlcNAcPRED-II’s independent test set are presented in Table 3.

Building upon the findings in Table 2, it can be inferred from Table 3 that LM-OGlcNAc-Site exhibited comparatively better performance in most of the evaluation metrics of O-GlcNAcPred-II’s independent test set.

### 2.1. Comparison with Existing Tools

For a fair comparison of the proposed methods with the existing predictors, we trained the three different FFNN architectures using Ankh, ESM-2, and ProtT5 embeddings on O-GlcNAcPRED-II’s [20] training data and evaluated performance on their independent test set. The training dataset contained 889 positive sites and 48,262 negative sites. The independent test set consisted of 357 positive and 25,093 negative sites. Due to the unavailability of O-GlcNAcPRED-II’s complete dataset and updates of some sequences in UniProt database, we were unable to match a few of the proteins, which led to a small variation in numbers. The results of the three FFNN models trained on Ankh, ESM-2, and ProtT5 embeddings of O-GlcNAcPRED-II’s training dataset and tested against O-GlcNAcPRED-II’s independent test set are shown in Table 4.

It can be observed from the table that almost all individual models and other combined models performed better than O-GlcNAcPRED-II. Furthermore, we can observe that our proposed decision-level fusion model (LM-OglcNAc-Site) performed better when compared to other combinations.

The table indicates that almost all individual models and our proposed combined model outperformed O-GlcNAcPRED-II [20], which is widely regarded as a state-of-the-art method. Notably, our proposed decision-level fusion model (LM-OglcNAc-Site) exhibited even better performance than all other combinations of models, including O-GlcNAcPRED-II [20]. Furthermore, to facilitate a better understanding of the comparison, Figure 1 presents radar plots depicting the sensitivity, specificity, MCC, and ROC-AUC values obtained from various models (such as pLM-FFNN, score-level fusion, and decision-level fusion). In the radar plot, points that reach further towards the edge of the spoke indicate higher values. Figure 1a,b show the results of individual models trained on Ankh, ESM-2, and ProtT5 embeddings and other combined models tested against our independent test and O-GlcNAcPRED-II’s independent test set, respectively. Looking at both plots in Figure 1, it is evident that LM-OglcNAc-Site had the highest values in most of the compared metrics.

We also performed similar experiments to those presented in Table 2 and Table 3 using the cost-sensitive method, where we used imbalanced number of positive and negative samples to train the model (refer to Appendix A). The corresponding results from the cost-sensitive methods are presented in Appendix A.

### 2.2. Case Study: Predicted Sites for O-GlcNAcylation in Human Galectins

Furthermore, we assessed the efficiency of our LM-OGlcNAc-Site tool in predicting O-GlcNAcylation sites in human galectins, which were previously reported as predicted by OGTSite and YinOYang in silico analysis [37]. Our results showed that there were multiple overlaps between all tools. Additionally, LM-OglcNAc-Site identified several new sites which can be potentially O-GlcNAcylated (Table 5).

Moreover, we evaluated LM-OGlcNAc-Site’s effectiveness in identifying sites located within intrinsically disordered regions (IDRs), as identified by the flDPnn tool [38]. Our findings indicate that our tool is capable of effectively detecting sites occurring within these regions. (Refer to Appendix A for more details.)

## 3. Discussion

This work provides a comparative analysis of three different pre-trained protein language models (Ankh, ESM-2, and ProtT5) for protein *O*-GlcNAc site prediction. In order to perform the analysis, we developed a framework for the prediction of protein *O*-GlcNAc sites using embeddings from pre-trained protein language models. Our framework consisted of learning from these *O*-GlcNAc site labels for supervised training using an artificial neural network. Additionally, to leverage the embeddings from these three protein language models, we performed integration of the embeddings using score- and decision-level fusion. Our results indicate that the decision-level fusion of the embeddings of these three protein language models is particularly useful for the prediction of protein *O*-GlcNAc sites.

As can be observed from the results, LM-OGlcNAC-Site is particularly able to perform better by integrating the embeddings from the individual protein language models. In summary, the results of this tool showcase the immense potential of integrating embeddings from multiple protein language models, enabling researchers and practitioners to delve into the intricate realm of post-translational modifications. The improvement in the results could be attributed to the fact that the three language models have different underlying transformer architectures and that the dataset used to train these models varied slightly, resulting in complementary representations. To the best of our knowledge, this work is one of the first works that integrates embeddings from multiple protein language models for the prediction of post-translational modification tasks. One of the possible drawbacks of the approach could be the computational overhead to extract these features and train the model to integrate these features.

## 4. Materials and Methods

### 4.1. Dataset

The dataset employed in this research was procured from O-GlcNAcAtlas (version 2.0), a dedicated repository of experimentally identified *O*-GlcNAc sites and proteins [12]. This collection comprises two discrete categories of data, distinguished by their identification clarity as either unambiguous or ambiguous *O*-GlcNAc sites. For the purposes of ensuring model reliability in this study, only unambiguously identified sites were taken as positive sites. After conducting a thorough process of duplicate removal and dataset cleaning, a total of 13,900 experimentally verified sites from 5355 proteins were obtained. Given the problem statement that necessitated a positive–negative learning approach, all other Serine (S) and Threonine (T) residues within these 5355 protein sequences that were not annotated as *O*-GlcNAc sites (including both ambiguously and unambiguously identified sites) were designated as negative sites. Subsequently, the entire dataset was partitioned in a 9:1 ratio, ensuring no overlapping of proteins, thus yielding a training set and a test set with 4826 and 529 sequences, respectively. Table 6 presents the overall structure of the dataset utilized in this study.

Given the uneven distribution of positive and negative samples within the training dataset, we employed a random under-sampling (RUS) technique on the negative set to address the issue of class imbalance. In addition to RUS, we also implemented imbalance learning using cost-sensitive (CS) learning. In this method, a higher penalty is assigned to misclassification errors for positive samples to counterbalance the overrepresentation of negative samples. The specifics regarding the dataset prepared using CS learning, and the results derived from it, are comprehensively detailed in Appendix A.

### 4.2. Sequence Encoding Using pLMs

In the domain of deep learning, the transformation of raw data into numeric space via feature vectors is crucial for robust computation and interpretation. This principle is particularly true when applied to raw protein sequences, where the quality of the feature representation plays a significant role in the overall performance of the model. Recently, the surge in popularity of transformer-based large language models in the field of natural language processing (NLP) has prompted the training of various analogous models on protein sequences. The aim of such endeavors is to harness the power of these transformer models to enhance the representation of proteins. These protein language models (pLMs) have the capacity to learn a contextualized representation (also called embeddings) from a vast corpus of protein sequences, an ability that has significant implications for downstream tasks related to protein functions. Using multi-head attention mechanisms and positional encoding, pLMs can effectively capture both localized and globalized context-sensitive features of protein sequences’ structures and functions.

In the context of PTM prediction problems, pLMs can be utilized to extract contextualized representations (or embeddings) for the site of interest (in this case, Serine (S) and Threonine (T) residues). With the rising popularity and demonstrated efficacy of these embeddings [23,39,40,41,42], this study explores three popular transformer-based protein language models, Ankh, ESM-2, and ProtT5, and their application in *O*-GlcNAc prediction tasks. Let *n* denote the length of a protein sequence and *L* be the embedding dimension (size of feature vector obtained per amino acid) from a pLM (Ankh/ESM-2/ProtT5). The resulting dimension for the entire sequence is *L* × *n*, while the dimension specific to the sites of interest, ‘S’ and ‘T’, is *L* × 1. Consequently, we use feature vectors, each with a length of *L* × 1, to contextually represent the positive and negative S/T residues. The process of extracting embeddings for the site of interest is shown in Figure 2. Detailed summaries of these pLMs, focusing on their role in this task, are provided in the subsequent sections.

#### 4.2.1. Ankh

Ankh [22] is a general-purpose protein language model that has a relatively smaller number of parameters as compared to other competitive protein language models. It was trained on a Uniref50 dataset consisting of 45 million protein sequences. The developers have released two versions of Ankh: Ankh_base and Ankh_large. Ankh_base is a lightweight version that has 450M parameters and which has an embedding dimension of 768, whereas Ankh_large has 1.15B parameters with an embedding dimension of 1536. In this work, we used Ankh_large model as an Ankh feature extractor. The model takes an overall sequence as input and returns a feature vector with a dimension of 1536 for each amino acid.

#### 4.2.2. ESM-2

Evolutionary Scale Modeling (ESM) [23] is a general-purpose protein language model based on the BERT transformer architecture and trained on UniRef50. ESM models are trained to predict masked amino acids using all the preceding and following amino acids in the sequence. There are different variants of models based on factors including number of parameters and dataset. In this work, we used a model called esm2_t36_3B_UR50D (hereafter termed ESM-2), which has approximately 3B learnable parameters. ESM-2 can take an input sequence with a length of up to 1024. Because of this sequence length restriction, we created a window of 1023 around the site of interest (511 residues on each side), and this window was used as input for the ESM-2 model. The output of the model was an embedding of a feature dimension of 2560 for each amino acid.

#### 4.2.3. ProtT5

ProtT5-XL-U50 [24] is a transformer-based pre-trained model trained on the top of Google’s T5 (Text-to-Text Transfer Transformer) [43] architecture on the BFD database and fine-tuned on the UniRef50 database. It consists of a 24-layer encoder–decoder with around 3B parameters. In this work, only the encoder part of the model, which takes a sequence as an input and outputs an embedding vector with a dimension of 1024 for each amino acid, was used for feature extraction. The dataset, the transformer architecture, and other details of these three protein language models are presented in Table 7.

### 4.3. Model Architecture

The overall architecture of our framework is composed of two levels. The first level (level-0) incorporates pre-trained protein language models (pLMs) that extract static global contextual embeddings of serine (S) and threonine (T) residues using the full sequence; each of these inputs has a dimension of *L* × 1, where *L* is 1536 for Ankh, 2560 for ESM-2, and 1024 for ProtT5. These *L* × 1 dimensional embeddings are then processed by a feed-forward neural network (FFNN) in the subsequent level (level-1) that classifies the membership status (positive or negative) of the site of interest (S/T). We independently used embeddings from three pre-trained pLMs (Ankh, ESM-2, and ProtT5) to acquire their respective classification probabilities. We refer to the suite of models consisting of each of these independently trained models of pLM embeddings as ‘pLM-FFNN’. This suite includes the Ankh-FFNN, ESM2-FFNN, and ProtT5-FFNN.

The underlying FFNN architectures in each pLM-FFNN consisted of hidden layers followed by a dropout layer to mitigate overfitting and an output layer with a single neuron. This architecture was determined through 10-fold cross-validation, utilizing different combinations of hyperparameters (including the number of layers, the number of neurons, and the dropout size) in a grid search procedure. We employed the ReLU activation function across all hidden layers and a sigmoid function in the output layer. Details regarding the hyperparameters and other configurations associated with Ankh-FFNN, ESM2-FFNN, and ProtT5-FFNN are presented in Appendix A Appendix A. In alignment with the findings of Villegas-Morcello et al. and Weissenow et al. [41], our work also demonstrates that pLM-based features can achieve competitive performance without requiring a complex architecture.

Additionally, we optimized the decision threshold cut-off for each of these models using threshold-moving strategies through 10-fold cross-validation based on the receiver operating characteristic (ROC) and precision-recall (PR) curves. As presented in Figure 3, we utilized the ROC curve to identify the optimal threshold value for each of the three pLM-based models by maximizing the geometric mean (g-mean) of sensitivity and specificity. The g-mean was computed at each threshold cut-off on the ROC curve, and the threshold yielding the highest g-mean was selected as the optimal threshold.

Furthermore, to leverage the diverse representations from these three different pLMs, we performed score-level fusion (based on an average strategy) and decision-level fusion (based on a hard-voting mechanism) of Ankh-FFNN, ESM2-FFNN, and ProtT5-FFNN models. This process allowed us to establish a framework that leveraged the strengths of the three different pLM-based protein sequence representations. A high-level depiction of this proposed architecture is provided in Figure 4.

### 4.4. Model Training and Evaluation

The parameters across all the pLM-FFNN models were optimized using the Adam optimizer, based on binary cross-entropy loss or log loss, with an adaptive learning rate set to 10^−5^. Prior to training, the batch size was set to 32 and the number of epochs was set to 100. To track the progress in loss reduction and to prevent overfitting, an early stopping strategy was used with a patience equal to 5.

Next, to assess and compare the performances of different models, we used a standard binary classification metric, the confusion matrix. The matrix consisted of four parameters: True Positives (TPs), indicating correct predictions of positive sites as *O*-GlcNAc sites; True Negatives (TNs), denoting correct predictions of negative sites as non-*O*-GlcNAc sites; False Positives (FPs), representing incorrect predictions of negative sites as *O*-GlcNAc sites; and False Negatives (FNs), marking incorrect predictions of positive sites as non-*O*-GlcNAc sites. With the use of these components, we were able to determine key evaluation metrics for each experiment, such as the Accuracy (ACC), Matthew’s Correlation Coefficient (MCC), Sensitivity (Sn), and Specificity (Sp). More elaborate explanations and the equations used for these metrics can be found in Appendix A Appendix A. Additionally, to better gauge the discriminative capacity of the models, we computed the area under the receiver operating characteristic curve (AUROC).

## 5. Conclusions

In conclusion, our work underscores the significance of a framework that uses a neural network architecture for integrating embeddings from multiple protein language models and high-quality datasets and effective predictive modeling techniques, resulting in the creation of LM-OGlcNAc-Site, a new predictor to identify *O*-GlcNAc sites in given protein sequences.

By training a feed-forward neural network with features from three protein language models (Ankh, ESM-2, and ProtT5) and integrating the embeddings from these three language models, our tool achieved performance improvement in predicting *O*-GlcNAc modifications. Leveraging the strengths of multiple protein language models, LM-OGlcNAc-Site outperforms existing *O*-GlcNAc prediction tools, demonstrating its superior performance. This development highlights the vast potential of protein language models, empowering researchers to explore the usage of these models to study post-translational modifications and other downstream bioinformatics tasks.

## Figures and Tables

**Figure 1 ijms-24-16000-f001:**
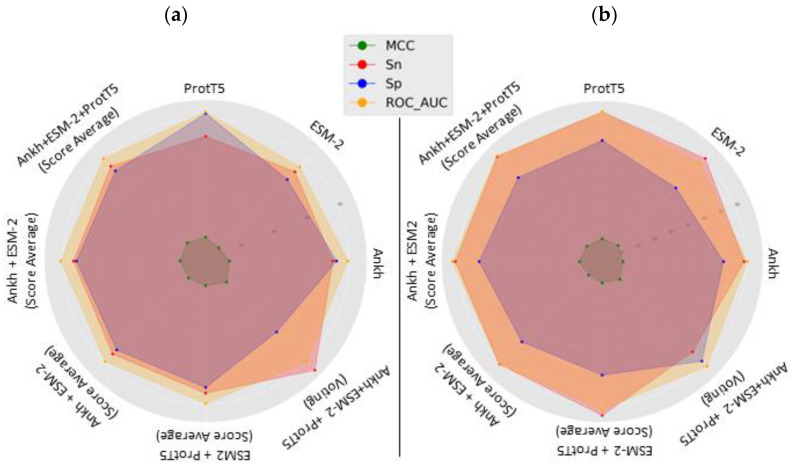
Radar plots to visualize performance evaluation of (**a**) results of our independent test set, as shown in Table 2, and (**b**) results of our independent test set, as shown in Table 3.

**Figure 2 ijms-24-16000-f002:**
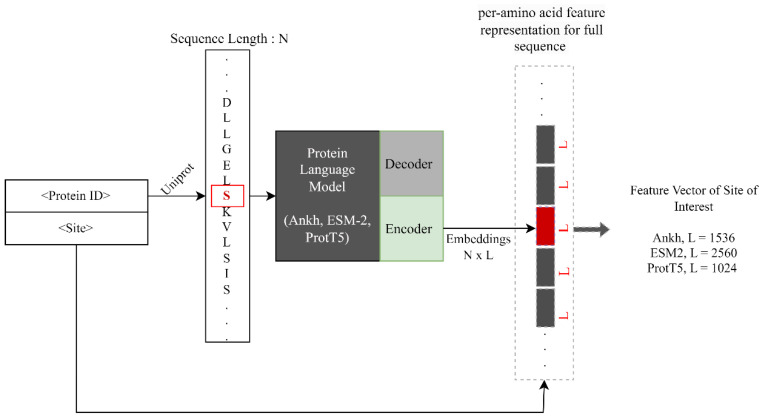
A high-level view of the process of extracting embeddings per amino acid using protein language models. The residue ‘S’ in red box shows the site of interest. The overall protein sequence is provided as an input to these three pLMs and the feature vector of corresponding sizes are obtained for the site-of-interest depending upon the pLM.

**Figure 3 ijms-24-16000-f003:**
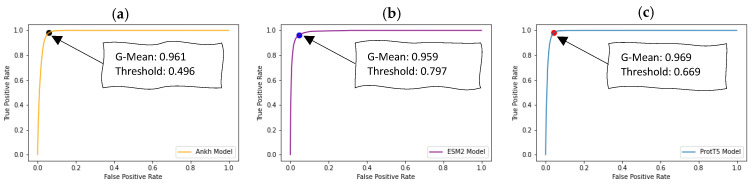
Determination of the optimal probability threshold cut-off that separates the positive and negative classes using ROC curve by maximizing g-mean of specificity and sensitivity. The figure illustrates the optimal cut-off points determined from training dataset using (**a**) Ankh embeddings, (**b**) ESM-2 embeddings, and (**c**) ProtT5 embeddings.

**Figure 4 ijms-24-16000-f004:**
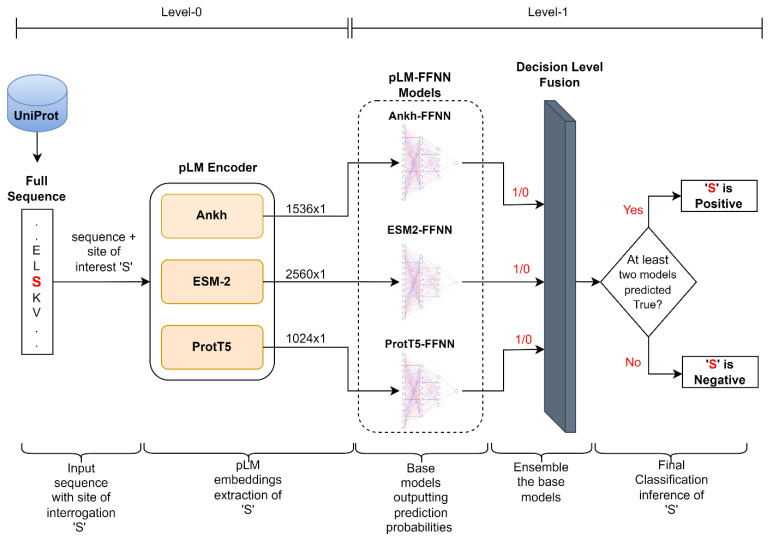
Diagrammatic illustration of the proposed model architecture. The input protein sequence of length *n*, with ‘S’ (highlighted in red) as the site of interest, is processed through encoders of three pLMs. Each pLM generates contextual embeddings of the sequence, resulting in a size of *L* × *n*. However, only vectors of size *L* × 1, corresponding to ‘S’ (site of interest), are extracted from these pLMs. These vectors serve as input to the pLM-FFNN models, where each model classifies whether ‘S’ is positive (1) or not (0). Finally, a decision-level fusion of the individual FFNN models is performed to obtain the final prediction.

**Table 1 ijms-24-16000-t001:** Results of 10-old cross-validation (mean and one standard deviation) of three FFNN models trained on Ankh, ESM-2, and ProtT5 embeddings. A total of five performance metrices, including accuracy, sensitivity, specificity, MCC (Matthew’s correlation coefficient), ROC-AUC (receiver operating curve–area under the curve), are reported.

Model	Mean Accuracy	Mean Sensitivity	Mean Specificity	Mean MCC	Mean ROC-AUC
**Ankh-FFNN**	0.7141 (0.0078)	0.7236 (0.0267)	0.7047 (0.0287)	0.4289 (0.0152)	0.7915 (0.0104)
**ESM2-FFNN**	0.7026 (0.0130)	0.7497 (0.0480)	0.6556 (0.0615)	0.4092 (0.0235)	0.7530 (0.0243)
**ProtT5-FFNN**	0.7152 (0.0068)	0.7352 (0.0073)	0.6952 (0.0149)	0.4308 (0.0134)	0.7872 (0.0105)

Values in parenthesis represent one standard deviation: (one S.D.).

**Table 2 ijms-24-16000-t002:** Results of three FFNN models trained on Ankh, ESM-2, and ProtT5 embeddings of training set and tested against independent test set. The bolded numbers show the highest values of each column.

Model	Feature Type	Accuracy	Sensitivity	Specificity	MCC	ROC-AUC
Individual Models	Ankh	0.7126	0.7122	0.7325	0.1329	0.7962
	ESM-2	0.7059	0.7071	0.6457	0.1050	0.7438
	ProtT5	0.6999	0.6981	**0.8263**	0.1334	**0.8348**
Score-Level Fusion	Ankh + ESM-2 + ProtT5	0.7495	0.7502	0.7141	0.1447	0.8116
	Ankh + ESM-2	0.7394	0.7397	0.7229	0.1423	0.8111
	Ankh + ESM-2	0.7358	0.7364	0.7038	0.1350	0.7954
	ESM-2 + ProtT5	0.7396	0.7402	0.7078	0.1379	0.7985
Decision-Level Fusion (LM-OGlcNAc-Site)	Ankh + ESM-2 + ProtT5	**0.8590**	**0.8648**	0.5613	**0.1659**	0.8116

**Table 3 ijms-24-16000-t003:** Results of three FFNN models trained on our training data using Ankh, ESM-2, and ProtT5 embeddings and tested against O-GlcNAcPRED-II independent test set. The bolded numbers show the highest values of each column.

Model	Feature Type	Accuracy	Sensitivity	Specificity	MCC	ROC-AUC
Individual Model	Ankh	0.6767	0.7899	0.6751	0.1161	0.8077
	ESM-2	0.5833	0.8123	0.5800	0.1270	0.7717
	ProtT5	0.6765	0.8347	0.6742	0.1270	0.8348
Score-Level Fusion	Ankh + ESM-2 + ProtT5	0.6634	0.8263	0.6611	0.1205	0.8320
	Ankh + ESM-2	0.6871	0.8179	0.6852	0.1266	**0.8350**
	Ankh + ESM-2	0.6350	0.8095	0.6326	0.1074	0.8117
	ESM-2 + ProtT5	0.6354	**0.8571**	0.6323	0.1189	0.8258
Decision-Level Fusion (LM-OGlcNAc-Site)	Ankh + ESM-2 + ProtT5	**0.7836**	0.7115	**0.7846**	**0.1403**	0.8320

**Table 4 ijms-24-16000-t004:** Comparison of our various models with the existing state-of-the-art model (O-GlcNAcPRED-II). All the models were trained on O-GlcNAcPRED-II’s training dataset. The highest values are bolded in each column.

Model	Feature Type	Accuracy	Sensitivity	Specificity	MCC	ROC-AUC
Individual Model	Ankh	0.7548	0.6162	0.7568	0.1015	0.7468
	ESM-2	0.6834	0.7199	0.6829	0.1014	0.7561
	ProtT5	0.7263	0.7086	0.7265	0.1140	0.7707
Score-Level Fusion	Ankh + ESM-2 + ProtT5	0.7367	0.7148	0.7371	0.1189	0.7892
	Ankh + ESM-2	0.7232	0.7030	0.7235	0.1114	0.7812
	Ankh + ESM-2	0.7314	0.6946	0.7319	0.1124	0.7820
	ESM-2 + ProtT5	0.7314	0.7254	0.7275	0.1188	0.7803
Decision-Level Fusion (LM-OGlcNAc-Site)	Ankh + ESM-2 + ProtT5	**0.8695**	0.5042	**0.8747**	**0.1322**	**0.7892**
O-GlcNAcPRED-II [20]		0.7239	**0.6712**	0.7246	0.1012	0.7433

**Table 5 ijms-24-16000-t005:** Predicted sites for O-GlcNAcylation in human galectins.

Galectin	NCBI Reference Sequence	OGTSite	YinOYang	LM-OGlcNAc-Site
Galectin-1	NP_002296.1	S84	S84	T71, S84
Galectin-2	NP_006489.1	S23, S51, S80, T85, S122	no predictions	T21, S80, T85
Galectin-3	NP_002297.2	S84, T133	S84, S91, S92, T98, T104, T243	S6, S12, S14, S40, S84, S91, S92, S96, T98, T104
Galectin-4	NP_006140.1	S40	S58, T217, T317	S86, S92, T217, S258
Galectin-7	NP_002298.1	S9	S2, S8, S9, S45, T57, T58	S2, T18, S31, S69
Galectin-8	NP_006490.3	T22, S152, T160, T201, T211	T160, S178, S358	S4, T160, S171, S178, T180, S192, T198, T201, S203, T207, T211, T215
Galectin-9	NP_033665.1	S18, T32, T152, S165, T193, S202	S4, S6, S12, S18, S139, T152, S160, T161, S165, T193, T195, T351, T355	S4, S6, T152, S160, S161, S165, T193, T195
Galectin-10	NP_001819.2	T9	T9, T16	S13
Galectin-12	NP_001136007.2	T81, T82, S143, S192, S221, T232, S305	T82, S192, T232, S315	S2, S221
Galectin-13	NP_037400.1	S13, S119, S127, T133	S2, S3, S13, S119, S127, T133	S2, S13
Galectin-14	NP_064514.1	T9, S13	S2, S3, T9, S13, S138	S2, S13, S138
Galectin-16	NP_001177370.2	S13, S119	S13, S119	S2, S13

**Table 6 ijms-24-16000-t006:** The number of experimentally verified positive (*O*-GlcNAc sites) and negative sites (before under-sampling) and train–test split of the dataset.

	No. of Proteins	Positive Sites	Negative Sites	
**Train**	4826	12,644	Before Under-Sampling	662,081
			After Under-Sampling	12,644
**Test**	529	1256	64,927	

**Table 7 ijms-24-16000-t007:** Description of the three protein language models used in the study.

pLM	Dataset	Architecture	Developer	Number of Parameters (Billions)	Embedding Dimension Per Residue (L × 1)
Ankh [22]	Uniref50 [44]	ConvBERT	Rostlab	1.2B	1536 × 1
ESM-2 (esm2_t36_3B_UR50D) [23]		RoBERTa	Meta	3B	2560 × 1
ProtT5 (ProtT5-XL-U50) [24]		T5	Rostlab	3B	1024 × 1

## Data Availability

The datasets, source code, and other relevant information are available at the GitHub repository at https://github.com/KCLabMTU/LM-OGlcNAc-Site (accessed on 5 November 2023). The webserver for LM-OGlcNAc-Site is available at http://kcdukkalab.org/LMOGlcNAcSite (accessed on 5 Novermber 2023).

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
