# Peer review of "Integrating Embeddings from Multiple Protein Language Models to Improve Protein O-GlcNAc Site Prediction"

_ijms, 2023, doi:10.3390/ijms242116000_

Round 1

Reviewer 1 Report

Comments and Suggestions for Authors

 The authors have developed a robust method for global and site specific O-GlcNAcylation prediction with high accuracy and sensitivity. Techniques for any types of O-GlcNacylation analysis are always important and essential for the development of the field. Thus this work is important and can be publish after some minor changes as suggested below.

1. All 'O' in O-linked, 'N' in N-acetyl and "O' in O-GlcNAc should be in italics.

2. Page 1 line 2-3: "Distinct from other glycan....is a unique....... monosaccharide modification" Please write 2-3 sentences how is this unique and its importance.

3. Page 2: "Prediction of O-Glycosylation site..." was it suppose to be 'O-GlcNAcylation'? I'm confused.

4. Add this reference "Olivier-Van Stichelen, S., Malard, F., Berendt, R., Wulff-Fuentes, E. & Danner, L. Find out if your protein is O-GlcNAc modified: The O-GlcNAc database. The FASEB Journal 36 (2022)."

5. Page 3; "additionally, the models have.... class differentiation" The sentence too too big and difficult to understand, please split this into simple sentences.

6. Table 4: Bold the higher sensitivity value 0.6712

Comments on the Quality of English Language

Some of the sentences are complex and too long making the whole statement difficult to digest.

Reviewer 2 Report

Comments and Suggestions for Authors

The manuscript entitled “Integrating Embeddings from Multiple Protein Language Models to Improve Protein O-GlcNAc Site Prediction” presents a new platform “LM-OGlcNAc-Site” for predicting O-GlcNAc modification on proteins. The authors highlighted the advantage of leveraging different protein sequence-based large protein language models (pLMs) to achieve higher sensitivity and accuracy of O-GlcNAc site prediction, showing the potential of implementing the “decision level fusion” in predicting other types of post-translation modifications.

Overall, the authors provide an improved in silico method to study O-GlcNAcylation that is known for its diversity and complexity in targeting protein sequences. The manuscript could be further improved if the following comments could be appropriately addressed:

Major points:

1. In Table 2, LM-OGlcNAc-Site showed superior performance in terms of accuracy and sensitivity compared to other modules including individual or score level fusion. However, its “specificity” value was significantly lower (0.56) compared to others (0.65-0.83). I would expect that the authors could provide a possible explanation about this decreased specificity and address how this would affect the O-GlcNAc site prediction.

2. One of the major challenges in O-GlcNAc sequence analysis is that the O-GlcNAcylation commonly occurs within the disordered or Ser/Thr-rch regions (e.g. C-terminal domain of RNA polymerase II or nuclear pore glycoprotein 62), which greatly increases the difficulty to unambiguously assign the O-GlcNAc localization. It would be of great importance to show the performance of LM-OGlcNAc-Site particularly in internal disordered regions (IDRs) or Ser/Thr-rich areas, either using case studies or small-scale analysis compared to other prediction tools.

3. Regarding the long history of O-GlcNAc site-detection technique development, a high number of experimentally detected O-GlcNAc sites were discovered under the approach of metabolic labeling with sugar analogues, which was recently showed to introduce false-positive identification of many O-GlcNAc sites (Angew Chem Int Ed Engl. 2018 Feb 12;57(7):1817-1820). It would be necessary to specify in the methods or other sections whether the potential false-positive O-GlcNAc information was excluded from the training dataset.  

4. From both the manuscript and website LM-OGlcNAc-Site, there is no description about whether the length of input sequence, such as inputs using full-length or partial sequence of RNA polymerase II, could affect the prediction outcome of LM-OGlcNAc-Site. Since different sequence lengths are expected to change the overall sequence composition and the number of O-GlcNAc site candidates, it would be important to clarify or discuss this issue in the manuscript.

Minor points:

1. In line 17 of the Abstract, “post translation modification” should be “post-translational modification”.

2. In line 7 of the 3rd paragraph and line 3 of the 5th paragraph of Introduction, additional space should be deleted after “PGlcS [16] ” and “protein language models [21–25] “, respectively.

3. In Results, “Supplementary Information Table: S3” should be “Supplementary Information Table S3”. 

4. For some of the abbreviations used in the Results (e.g. MCC, ROC), their full names were described later in the section Materials and Methods. It could be better to move the full-name annotation of the abbreviations to the sites where they were first mentioned.

5. It would be more reader-friendly to rearrange the order of Supplementary Tables based on their sequence of being mentioned in the manuscript including the Results and Materials and Methods.

Reviewer 3 Report

Comments and Suggestions for Authors

This article addresses an important question about prediction of sites for O-GlcNAcylation in proteins. This topic is very interesting, and the authors examine the performance of three so-called language models in this context. I cannot review the results because they are outside of my expertise. However, I think the article would be more informative if the authors provide an example of their analysis with respect to specific proteins, which were already predicted to be O-GlcNAcylated using other well-known prediction programs such as YinOYang and OGTSite (e.g. reported for galectins in PMID: 32731422). As a minor comment, the article has multiple technical issues with many references, which contain missing citation info: 11, 13, 17, 21, 23-26, 33, 35, 36, 40, 41.

Round 2

Reviewer 2 Report

Comments and Suggestions for Authors

The manuscript has been revised nicely. No further comments/suggestions.

Reviewer 3 Report

Comments and Suggestions for Authors

Consider to be clear about results presented in the Table 5. This reviewer suggests the following revision of the relevant portion of the text:

"Furthermore, we assessed the efficiency of our LM-OGlcNAc-Site tool in predicting O-GlcNAcylation sites in human galectins, which were previously reported as predicted by OGTSite and YinOYang in-silico analysis [37]. Our results showed that there were multiple overlaps between all tools and, additionally, LM-OglcNAc-Site identified several new sites, which can be potentially O-GlcNAcylated (Table 5)."

Comments on the Quality of English Language

The authors addressed my comments.
